# Applying the FFP Approach to Wider Land Management Functions

**Kathrine Kelm** [1], **Sarah Antos** [1] **and Robin McLaren** [2,*]

1    Urban, Disaster Risk Management, Resilience & Land, World Bank, Washington, DC 20433, USA;
     kkelm@worldbank.org (K.K.); santos1@worldbankgroup.org (S.A.)
2    Know Edge Ltd., Edinburgh EH14 1AY, UK
*    Correspondence: robin.mclaren@KnowEdge.com; Tel.: +44-78-03163137

**Abstract:** The initial focus of implementing the Fit-for-Purpose Land Administration (FFPLA) methodology was to address the significant, global security of tenure divide. We argue that this land tenure methodology is proving successful in scaling up the provision of security of tenure for developing countries. The increasing adoption of the FFPLA methodology has also opened opportunities and provided flexibility for the innovative use of emerging technologies to accelerate the global roll out of security of tenure, such as the use of autonomous drones and machine learning techniques applied to image analysis. Despite wider adoption of participatory approaches to the recording of land tenure, similar FFP solutions for the other components of land administration services (land value, land use and land development) and land management functions are still evolving. This article therefore explores how the FFP approach can be applied to this wider set of land administration services and land management functions. A case study methodology, using three case studies, is used to determine if the case study approaches meet the FFP criteria. The focus is on the urban environment, drawing mostly from experiences and case studies in the Urban, Disaster Risk Management, Resilience & Land Global Practice of the World Bank. These opportunities for the wider application of the FFP approach and associated principles are being triggered by the innovative use of emerging new data capture technology developments. The paper examines the innovative use of these emerging technologies to identify a common set of data capture techniques and geospatial data that can be shared across a range of urban land administration and management activities. Finally, the paper discusses how individual land projects could be integrated into a more holistic land administration and management program approach and deliver a significant set of socio-economic benefits more quickly. It is found that the FFP approach can be more widely adopted across land administration and land management and in many cases can share a common set of geospatial data. The authors argue that the wider adoption and integration of these new, innovative FFP urban management approaches will require a significant cultural, professional, and institutional change from all stakeholders. Future work will explore more deeply these institutional weaknesses, which will provide a basis for guidance to the World Bank and similar institutions.

**Keywords:** fit-for-purpose land management; innovative technology; aerial and street level imagery; machine learning; integrated land programs

## 1. Introduction

### 1.1. Significance of Land Administration to the Sustainable Development Goals (SDGs)

The World Bank provides an excellent example of the significant role land administration has in supporting the Sustainable Development Goals (SDGs). In 2013 the World Bank adopted the Twin Goals to: End Extreme Poverty—Reduce the percentage of people living on less than USD 1.90 a day to 3 percent by 2030; and Promote Shared Prosperity—Foster income growth of bottom 40 percent of the population in every country. Achieving the Goals in a Sustainable Manner- Securing the long-term future of the planet and its

resources, ensuring social inclusion, and limiting the economic burdens on future generations underpin efforts to achieve the two goals. The goals are aligned with the 2030 Sustainable Development Agenda, and the principles and targets embodied in the SDGs. Land, as a natural resource and strategic asset, features prominently in the SDGs and land administration, especially in securing land rights, and thus performs a fundamental role in reducing poverty and promoting shared prosperity. The World Bank is the custodian agency for key SDG indicators on land tenure security:

- SDG 1 Poverty, Target: 1.4: By 2030, ensure that all men and women, in particular the poor and the vulnerable, have equal rights to economic resources, as well as access to basic services, ownership and control over land and other forms of property, inheritance, natural resources, appropriate new technology and financial services, including microfinance; and
- Indicator: C.2: Percentage of people with secure tenure rights to land (out of total adult population), with legally recognized documentation and who perceive their rights to land as secure, by sex and by type of tenure.

World Bank programs and projects provide financing, capacity building and technical support for land administration and land management investments.

### 1.2. The Emergence of FFPLA

The delivery of security of tenure to citizens across the globe, especially in developing countries, through land administration services has often been inefficient and its outreach remains limited; it is estimated that around 75 percent of the world's population do not have access to formal systems [1] to register and safeguard their land rights—the majority of these are the poor and the most vulnerable in society [2]. The underlying reasons for this limited success include: the continued influence of the colonial legacy on land administration institutions, inequalities in land distribution, and legal/regulatory frameworks; reluctance of land professionals to adopt a new information management role and allow 'barefoot surveyors' to use new, participatory technologies and lower specifications for the spatial accuracy of the capture and maintenance of land rights information; endemic corruption and the protection of the elite; significant lack of capacity at all levels; aversion to integrate informal land rights into the formal land administration system; and lack of sustained political support for reforms.

However, the new global agenda and associated targets around the SDGs have triggered a strong, positive response from global land professionals since land governance is on the critical path of delivering 10 of the SDG targets [3]. The land administration sector is experiencing a significant transition in the adoption of emerging technologies and new flexible, faster and cheaper approaches [4]. The previous straitjacket of high quality and strict conventions is being relaxed and the flexibility of a continuum of approaches is increasingly being observed in literature and practice [5–8]. These new approaches are being based and designed around Fit-For-Purpose Land Administration (FFPLA) solutions that reflect the characteristics of being flexible, inclusive, participatory, affordable, reliable, attainable, and upgradeable. Cost effective and sustainable solutions can then be achieved in a relatively short timeframe [2]. This approach was defined originally by the International Federation of Surveyors (FIG) and the World Bank [2] and subsequently by a more detailed country implementation guide supported by the Global Land Tool Network [9]. Nationwide programs have included: the rural land certification programs of Ethiopia [7], Rwanda's Land Tenure Regularization (LTR) program [10], Kadaster International's fast and effective land administration program supporting the implementation of the Colombian Peace Agreements [11], a program to accelerate land rights redefinition and reconstruction in Nepal after the 2015 disastrous earthquakes [12], and the World Bank funded program to support the registration of all land parcels in Indonesia by 2025 [13]. These exemplary, mostly nationwide programs clearly reflect what FFPLA can achieve when there is top level government support.

### 1.3. New Innovative Technology Supporting FFPLA

One of the key drivers to support FFPLA, and a key element of participatory solutions, has been the inventive use of emerging, innovative technology [14]. A good example is the use of mobile phones by non-land professionals to capture land parcel boundaries and associated legal information to define, record and adjudicate land rights [15]. For example, USAID have embraced this technology to underpin their Mobile Application to Secure Tenure (MAST) project in Tanzania to improve land governance and lower the cost of land certification programs [16].

Unmanned Aerial Vehicles (UAVs) or drones are another game changing technology to significantly impact and accelerate the adoption of FFPLA approaches. Over 20 years ago, drones that started in military applications were adopted as hobby gadgets. However, they have quickly become applicable and very effective in commercial and scientific contexts. For example, as soon as regulations permit, Amazon plans to have a Prime Air Service that uses drones to deliver parcels up to five pounds (2.3 kg) in 30 min [17]. The delivery of life-saving medical supplies in adverse environments is increasingly feasible by drones [18]. Drones are also having a major impact on how geospatial information is being captured. These platforms are increasingly autonomous, able to work out of sight, and for long durations. They are also being supported by an increasingly rich portfolio of digital photogrammetry tools to flight plan and process in real-time or post-process. They can carry a wide range of sensors and are being increasingly used by both the public and private organizations and individuals. Drones can generate high-resolution imagery (often better than 10 cm resolution). Quality geospatial information can be derived, including orthoimages, digital elevation models and 3D point clouds. Drones have therefore created a complimentary niche between traditional ground surveying, aerial surveys and satellite imagery [19].

The costs of producing conventional aerial photographs, digital orthophotos and cadastral base maps often amount to a significant part of a project budget and there is often a considerable delay between data capture and delivery of the processed products. Drones offer a new approach, working at a smaller geographic coverage but producing high-resolution products as and when needed, thus ensuring that the information is current. As drone technology has become a tool adopted by local mapping companies and private surveyors, the time and cost to produce spatial data for land administration projects have dropped. The World Bank has concluded that embedding mapping capacity, including the use of drones, at the local level facilitates a FFP approach to respond more dynamically and economically to land tenure activities [20]. The World Bank was an early adopter and advocate of the use of drones to capture land parcel boundaries in a participatory approach within their land tenure programs. Drones were originally piloted in Albania (2014) then used in Kosovo (2016), Vietnam (2018) and the Philippines (2021).

Despite drones being relatively widely adopted in a range of applications [21], there are still some key issues that are currently inhibiting the greater use of this platform, including regulations [22] and safety concerns with the aviation sector, privacy, licensing, professional capacity, and pushback from land professionals over the perceived threat to their livelihoods. The use of drone and street level imagery has been further enhanced through the use of Machine Learning (ML) to automatically or semi-automatically extract objects from the imagery. The World Bank has developed ML solutions to extract information from georeferenced images [23]. So far, these solutions are being used for assessing building structures, including building materials, number of floors and windows, which can then be used for vulnerability assessment and disaster risk analysis, valuation, and other applications [23]. Research is maturing in the automatic extraction of land parcel boundaries from imagery to accelerate the mapping and registration of large numbers of unrecorded land rights globally [24]. Research has applied and compared the ability of rule-based systems within Object-Based Image Analysis (OBIA), as opposed to human analysis, to extract visible cadastral boundaries from very high-resolution World View-2 images, in both rural and urban settings [25].

### 1.4. Context of FFPLA in Land Governance

Land governance [26] is about sustainably and transparently managing land, property and natural resources. This is achieved through three fundamental components: Land Policies and strategies, Land Administration Services and Land Management Functions, and Land Information Infrastructures (Figure 1).

Comprehensive and robust land governance [27] depends upon a harmonized legal and regulatory framework, institutional structures with clear roles and responsibilities, and capacity to implement land related policies and strategies consistently across a country. Land administration services and associated land information infrastructure provide a country with the means to implement land policies and supporting land management strategies aligned with the SDGs. These land administration services include land tenure (securing and transferring rights in land and natural resources), land value (valuation and taxation of land and properties), land use (planning and control of the use of land and natural resources), and land development (implementing utilities, infrastructure, construction works, and urban and rural developments) [26]. These land administration services are fundamentally facilitated by access to a comprehensive land information infrastructure; one that consists of information on the built and natural (including marine) environment and is an integral part of a National Spatial Data Infrastructure (NSDI).

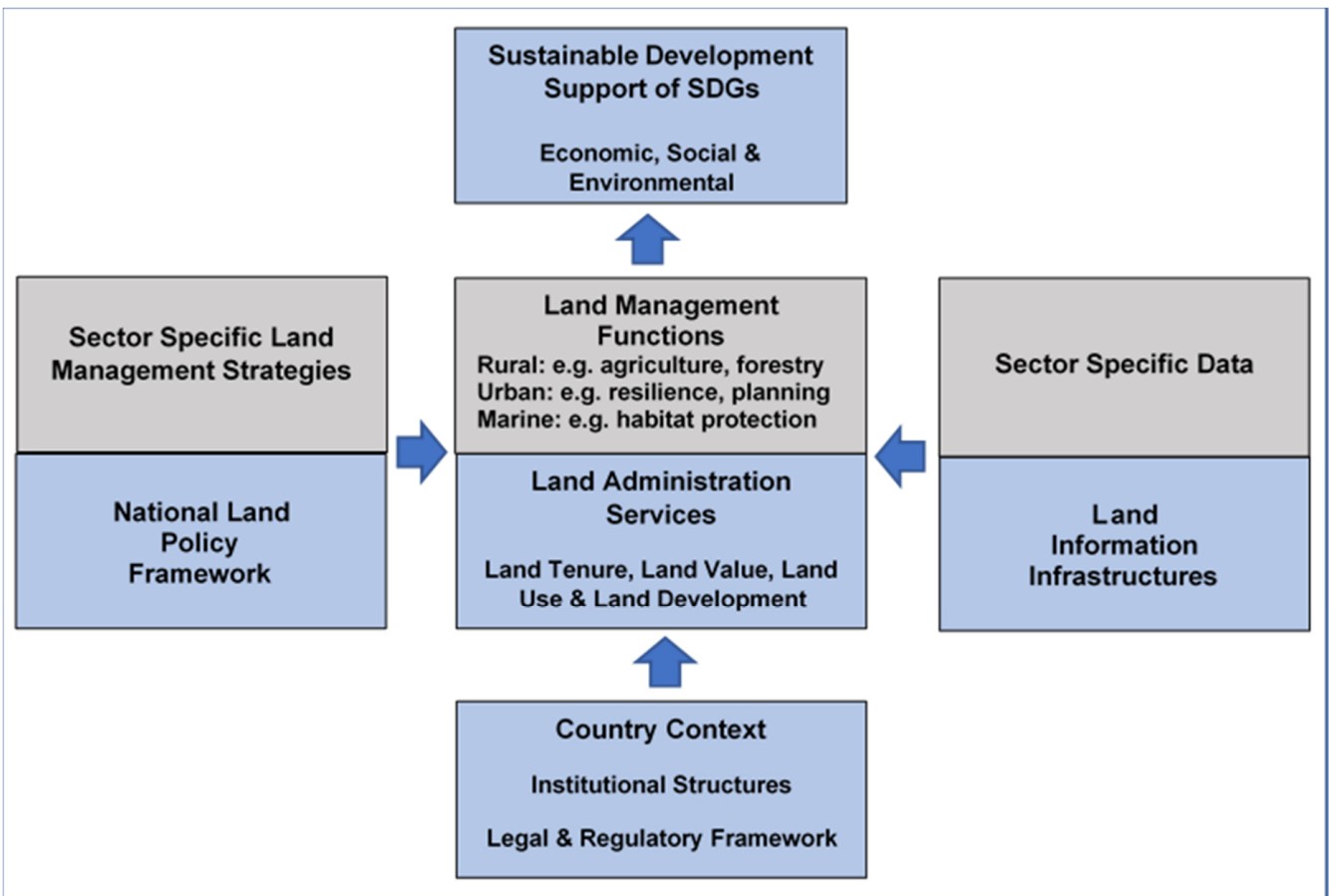

**Figure 1.** Land Governance Ecosystem (Adapted from The Land Management Paradigm in blue [28] (p. 171)).

Land management [28] (pp. 116–117) is primarily guided by an overall National Land Policy [29] that supports the formulation of sector specific land management strategies, such as forestry and agriculture in the rural context, city resilience and disaster risk management in the urban context, habitat protection and pollution control in the marine context, and climate change mitigation in an over-arching context. Sound land management functions ensures that decisions on land (rural, urban, and marine) are truly evidence based, using the

underpinning land administration services and associated land information infrastructures in conjunction with land sector specific data. This overall land governance ecosystem (Figure 1) ensures that sustainable development can be achieved.

Currently, the focus of applying the FFP paradigm [2] is land tenure. However, the use of innovative technologies is creating opportunities for the paradigm to be applied across a much broader set of land sector applications. Through case studies in the urban environment, this paper investigates how the FFP methodology can be applied to a wider set of land administration services and land management functions. The paper also evaluates if a common set of data capture techniques and geospatial data can be shared across a range of urban land administration and management activities. Finally, the paper discusses how individual land projects could be integrated into a more holistic land administration and management program approach and thus deliver a significant set of socio-economic benefits more quickly.

## 2. Approach and Methodology

The concept of the FFP approach emerged because the traditional approaches to land tenure projects were judged to be too expensive, took a long time to implement, the process was too rigid, and there were insufficient professional resources to implement and maintain at scale. Therefore, the FFP approach adopted the model of a Minimum Viable Product (MVP) where an entry level solution is developed that meets the basic needs of customers and one that can then be incrementally improved over time, where there is demand. In the case of FFP land tenure this involved reducing the accuracy specifications, using uncomplicated, participatory tools for data capture and being inclusive of all types of tenure.

The original definition of the FFP approach, outlined by FIG and the World Bank in 2014 [2], identified seven elements (Table 1) to be used in assessing the compliance of a proposed solution as being FFP.

**Table 1.** Characteristics of the FFP Approach [2].

| | |
|---|---|
| **Flexible** | in the spatial data capture approaches to provide for varying use and occupation. |
| **Inclusive** | in scope to cover all tenure and all land. |
| **Participatory** | in approach to data capture and use to ensure community support. |
| **Affordable** | or the government to establish and operate, and for society to use. |
| **Reliable** | in terms of information that is authoritative and up-to-date. |
| **Attainable** | to establish the system within a short timeframe and within available resources. |
| **Upgradeable** | with regard to incremental improvement over time in response to social and legal needs and emerging economic opportunities. |

These seven elements, detailed in Table 1, are used in the adopted methodology as a set of criteria to evaluate three urban case studies of new, innovative land administration services and land management functions to determine if they can be classified as FFP and used as cheaper and faster entry level solutions to manage the sustainable development of urban environments more effectively.

Wherever possible, the approaches used to support these new FFP solutions should mirror the tools and data being used for FFP land tenure solutions. This commonality will then allow more integrated and cost-effective FFP solutions to be created. Therefore, as a second step in the methodology, we evaluate the data capture technology and corresponding geospatial data requirements needed to support the FFP approach to land tenure against the three land administration services and land management functions case studies. This then enables us to determine if there is a common set of geospatial data that can be captured once and shared across many other land services and functions in the urban

environment. The starting assumption for the elements included in this common set of geospatial data are listed in Table 2:

**Table 2.** Elements of Common Set of Geospatial Data.

| |
|---|
| Drone imagery with a pixel resolution <10 cm from RGB digital camera and spatially corrected through Ground Control Points or Real-Time Kinematic (RTK). |
| Street level imagery with a resolution of 30 megapixels stitched together into georeferenced street panoramas and spatially corrected through Ground Control Points or RTK. |
| Digital Elevation Model/Digital Terrain Model (DEM/DTM). |
| 3D photogrammetric models to support data capture from the drone and street level imagery through digital photogrammetry. |
| 3D point cloud (either from the Structure from Motion (SfM) approach or using LiDAR). |
| Digital orthophoto. |

To effectively allow these data to be shared and support interoperability then all the data components must be compatible with international or national geospatial data standards, such as those published by the Open Geospatial Consortium [30].

This then leads to a discussion around the opportunity to have a more holistic approach to integrating the implementation of FFP land administration services and land management functions within an integrated land program rather than the current practice of implementing completely independent land projects.

The FFPLA concept is composed of three frameworks: spatial, legal and institutional. This study has primarily focused on the spatial framework and identified how emerging technologies are facilitating the wider use of the FFP approach. However, the successful implementation of these new FFP land administration services and FFP land management functions will require supportive changes to the legal and institutional frameworks. Although these non-technical transformations are not a core part of this article, they are important and are briefly included as part of the discussion of the article.

### 3. Emerging Innovative Technologies Supporting FFP Land Tenure

Disruptive and innovative technologies are being applied to the FFP land tenure approach to semi-automate elements of data capture, reduce the costs of obtaining imagery, lessen the need for highly skilled resources and provide high fidelity 3-D models. These technologies are reviewed here to determine the characteristics of their data outputs and therefore evaluate their applicability in other FFP land administration services and land management functions.

#### 3.1. Drones

Drones were not initially designed for professional geomatics and land administration applications. However, new technologies, including a wide range of miniaturized sensors (Table 3) have been developed for civilian application, opening new use opportunities. These enhancements have improved the flexibility (fewer restrictions in terms of a sensor's installation), performance (more duration, better aerodynamics profile, better navigation system), and planning tools (new tools have been developed for planning and control of the UAV operations) [31]. Fidelity has been revolutionized through the use of real-time RTK services and the emergence of 'RTK from the sky' [32]. The use of SfM applications that allow a 3D model to be derived from a sequence of images captured from different points of view [33] has further enhanced their use.

**Table 3.** Sensors Onboard Drones: Auxiliary and Specific [21]. Reprinted with permission from the American Society for Photogrammetry & Remote Sensing, Bethesda, Maryland, www.asprs.org/, accessed on 6 January 2021.

| Auxiliary | Specific | |
|---|---|---|
| • Ultraviolet spectrometer. | • Thermal cameras. | • Ultraviolet spectrometer. |
| • Multi-gas detector. | • Infrared cameras. | • Multi-gas detector. |
| • Sonar. | • FLIR. | • Sonar. |
| • Smartphone. | • LIDAR (Laser scanner). | • Smartphone. |
| • Aerosol sampling. | • Irradiance. | • Photometer, aethalometer. |
| • Pyranometer. | • Radar/SAR. | • Aerosol sampling. |
| • Particle counters (optical, condensation). | • Multi-Hyperspectral (HyperUAS). | • Probes (temperature, humidity, pressure). |
| • Photometer, aethalometer. | • Radiometer (multi-frequency). | • Cloud droplet spectrometer. |
| • Probes (temperature, humidity, pressure). | • Infrared spectroscopy. | • Pyranometer. |
| | • Video cameras (visible spectrum): EOS, stereoscopic, omnidirectional, fisheye lens. | • Particle counters (optical, condensation). |
| • Cloud droplet spectrometer. | | • Electrostatic collector. |
| | • VCSEL. | • Magnetic sensor. |
| | • WMS. | • Ultraviolet flame detector. |
| | • Electronic nose. | • Gas/smoke detector. |
| | | • Radiation gauge. |

There are several types of drones (Table 4) with various sizes and payloads. The majority of drones used for FFP land tenure have been in the nano and micro categories.

**Table 4.** Drone Classification [31].

| Category | Range (km) | Flight Height (m) | Duration (h) | MTOW [1] (kg) |
|---|---|---|---|---|
| Nano | <1 | <100 | <1 | <0.025 |
| Micro | <10 | 250 | 1 | <5 |
| Mini | <10 | 150–300 | <2 | 150 |
| Close range | 10–30 | 3000 | 2–4 | 150 |
| Short range | 30–70 | 3000 | 3–6 | 200 |

[1] Maximum Take-off Weight.

Various studies highlight the applicability of drones as an efficient tool for: (1) mapping customary land rights in Namibia [34]; (2) capturing cadastral boundary data in Indonesia [35]; and (3) Vietnam where UAVs produced survey grade orthophotos with positional accuracy compliant with existing surveying regulations [36]. In some cases, the legal and regulatory framework has been modified to accommodate a reduction in the surveying regulations and thus support the FFP methodology.

The typical outputs from the use of drones used in FFP land tenure applications are reflected in Table 2 and include:

- Imagery with a pixel resolution <10 cms from RGB digital camera;
- Spatially correct data through Ground Control Points/RTK post processing/Realtime RTK/RTK from Sky;
- DEM;
- 3D photogrammetric models to support data capture through digital photogrammetry;

- 3D point cloud (the SfM approach creates a low-density point cloud compared to the use of LiDAR);
- Digital orthophoto.

The key product derived from the drone to support the FFP land tenure methodology is the orthophoto. This is used to engage with citizens and communities to identify and record visual parcel boundaries. The orthophoto can be produced in paper form or used directly in digital form using mobile phones or tablets technologies. Machine Learning (ML) has been successfully used to extract parcel boundary objects [37,38] in order to better support the citizen engagement process.

Although drones can produce high-resolution imagery and derived 2D and 3D data very efficiently and cheaply with less skilled resources, there remain some constraints. For example, the duration of flights is limited by battery and payload capacity, and aeronautical regulations can constrain flying areas. Despite these restrictions, drones have become a game changer in efficiently capturing geospatial data over large areas.

### 3.2. LiDAR

LiDAR is a remote sensing technology, which emits lasers to collect measurements that can later be used to create 3D models and maps of objects and environments. LiDAR shoots pulses of light (up to a million pulses per second), which bounce off a surface and return to the sensor. The sensor then calculates the time it took for the ray of light to return and which direction it came from. This process ultimately creates a point cloud map of the scanned surroundings—reality capture [38]. LiDAR is increasingly being adopted by autonomous vehicles and this application is accelerating the research and capabilities of the technology. The key characteristics and value add of LiDAR are the inherent accuracy capability, the generation of a high degree of information completeness of reality and the ability to integrate a variety of additional sensor data that together provides a point cloud with high-fidelity visualization—highly accurate, high-definition and increased-density point cloud data with equidistant point patterns [39]. This allows Machine Learning and geospatial analytics to derive and extract spatial objects.

The typical outputs from the use of LiDAR to support FFP land tenure applications include:

- Spatially correct 3D point cloud with a high resolution (180 points/m$^2$) [40];
- DEM/DTM.

The key product derived from LiDAR to support the FFP land tenure methodology is the 3D point cloud that is used to derive the DEM, DTM and orthophoto, helps 3D visualization, and aids ML to extract objects, such as parcel boundaries [25]. LiDAR produces large volumes of accurate, consistent data quickly, but the technology is complex in nature and requires a deep understanding of the sensor and associated technical know-how.

### 3.3. Street Level Imagery

In addition to imagery being captured by airborne sensors, imagery from the street level is increasingly being used, for example, to support navigation and autonomous vehicles. Geo-located images taken by citizens and obtained at scale from 360-degree cameras, usually attached to cars and motorcycles, are stitched together to form interactive street panoramas. Google 'Street View', launched in 2007, has the most comprehensive coverage and by 2017 had captured more than 16 million kilometres of Google Street View imagery across 83 countries [41]. The other source of Google Street View imagery is from users. Google introduced this feature in 2017 to allow contributors to add their own images to the Google Street View database for possible inclusion into the 'map' [42]. The analysis of this imagery is highly restricted through licensing. Mapillary [43] and CartaView [44] (shared under a Creative Commons license) are other services sharing crowdsourced, geotagged street level images.

An example of this approach is a World Bank Global Program for Resilient Housing project in Padang, West Sumatra, Indonesia that captured drone and street-view imagery to create a building inventory for a study that supported a home improvement subsidy program, known as Bantuan Stimulan Perumahan Swadaya (BSPS). Normally, drones are perceived to be capable of just gathering information in small areas of about 2–5 km$^2$, depending on the spatial resolution, flight pattern and drone type. However, the project area of 90 km$^2$ was covered by drone and street-level imagery at a rate of 3 km$^2$ per day [23]. Street view imagery captured using a Trimble MX7 360 camera mounted on top of an SUV that has six, 5 megapixel cameras, the Trimble Applanix GNSS and inertial geo-referencing modules. This generated 30-megapixel geo-referenced images. Using the georeferenced, overlapping images it was possible to get robust object positioning, perform measurements within the images and create 3D models. The georeferenced street panorama can be feed into platforms like Mapillary for visualization and navigation, and also used to generate a housing stock inventory that includes ML derived characteristics such as building material, use, construction type, and vintage. This inventory in turn can be used to inform home improvement programs (see Figure 2), property valuation estimations, and disaster risk management exposure models, for example [23]. The typical outputs from the use of street level imagery include:

- Imagery stitched together into georeferenced street panoramas;
- Spatially correct data through RTK post processing/Realtime RTK;
- 3D photogrammetric models to support data capture through digital photogrammetry;
- Digital orthophoto.

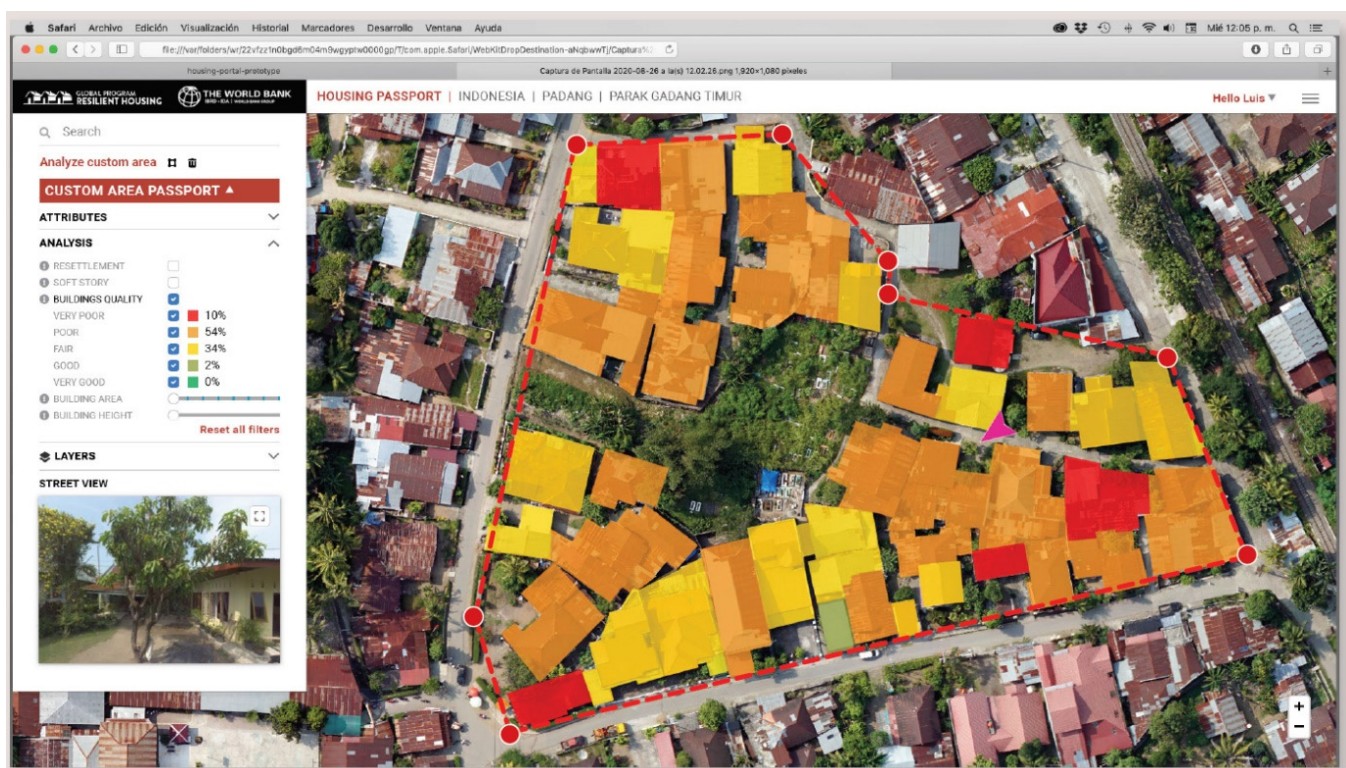

**Figure 2.** Results of building quality assessment, Penang, Indonesia [source World Bank].

The key product derived from street level imagery to support the FFP land tenure methodology is the georeferenced street panorama Although FFP land tenure can be completed using just drone imagery, the georeferenced street panorama derived from street view imagery can add further support to the FFP land tenure process. It can be used to perform measurements, help interpretation of the urban landscape and support allocation of street addresses. However, the street view imagery supports ML to extract building

characteristics from imagery [23] and consequently facilitates new FFP land administration services and FFP land management functions.

*3.4. Object Extraction from Imagery Using Machine Learning (ML)*

Artificial intelligence as defined by Moore 'is the science and engineering of making computers behave in ways that, until recently, we thought required human intelligence' [45]. However, ML is a branch of artificial intelligence, and as defined by Computer Scientist and machine learning pioneer Mitchell: 'Machine learning is the study of computer algorithms that allow computer programs to automatically improve through experience' [46]. There are three types of ML: supervised, unsupervised and reinforced. Supervised learning algorithms 'try to model relationship and dependencies between the target prediction output and the input features, such that we can predict the output values for new data based on those relationships, which it has learned from previous datasets' [47]. This type of ML is the most commonly used to support the extraction of features from imagery datasets to support land administration services and land management functions. For example, windows, doors and materials of buildings can be identified from street-level imagery and recorded to support the valuation of buildings once the algorithms have been trained through examples [23]. ML is playing an increasingly important role in the automatic capture of spatial data—particularly in the context of citizen science projects. For example, Varas-Munoz et al. [48] demonstrate the benefit of using convoluted neural networks (CNN) to support what is currently a human centered approach to creating content for OpenStreetMap. They review CNN techniques that have been used to automatically extract roads from imagery, and to identify geometric and semantic errors in the data entry process. The ambition is to incorporate ML techniques into various editing suits in order to increase both the speed and quality of data captured.

Cadastral intelligence [49] is the ability to apply spatial intelligence [50] to identify visible cadastral parcel boundaries. This is a key principle supporting approaches to FFP land tenure where citizens are provided with orthophotomaps to identify their properties and delineate the parcel boundaries. This is most successful where the parcel boundaries align or overlap with visual or topographic objects. However, to accelerate this process, ML is now being applied to extract cadastral parcel boundaries automatically from imagery and point clouds, and automated cadastral intelligence [25] is now being delivered. Significant research is taking place within the geospatial information and land administration domains to achieve this holy grail. Although tests have achieved around 50 percent success in rural areas, with poorer results in urban areas [49], the use of point clouds in the analysis could significantly improve this success rate [37]. However, even these initial, relatively low rates of automatic parcel boundary extraction will support and accelerate the current process of humans visually identifying the parcel boundaries.

The Technology & Innovation (TI) Lab of the World Bank have investigated the feasibility of using ML to automate the extraction of parcel features for parcel mapping procedures in Punjab, Pakistan. Three methods were prototyped: segmented and classified image using Support Vector Machine (supervised machine learning methods used for classification, regression analysis and outlier detection); edge detection; and segmented and classified image, using Deep Learning (a subset of ML that is data-driven modelling, which leverages the use of Neural Networks). The most promising results were obtained through the Deep Learning method [24] where there is significant research interest in this state of the art approach.

## 4. FFP Assessment of Wider Land Administration and Management Opportunities

The review of emerging innovative technology has identified that the technology is supporting and accelerating the adoption of FFP land tenure through data capture automation and acquisition of low-cost imagery, for example. However, can these tools and the associated data be directly applied to a wider set of land administration services and land management functions where the key FFP principles (Table 1) can be applied? Three

case studies from across the land sector, sourced from the World Bank, are investigated to review the opportunities of a new and wider set of FFP solutions that can share common geospatial data.

Although these case study projects have been directly managed, financed and often resourced by the World Bank, there is a clear objective of the World Bank to allow countries, especially low-income countries, to adopt these approaches without the direct support of the World Bank. This will allow countries to effectively manage private sector companies to capture the drone and the street level imagery and to feel comfortable to hire and oversee Machine Learning experts to analyze the imagery. This greater accessibility will then allow more countries to take advantage of these new approaches and create higher degrees of sustainability of their projects. This is already happening in countries like Colombia and Indonesia.

*4.1. Case Study 1: Valuation and Property Tax Revenues*

One of the main sources of revenue for Local Government is through property taxation. There are many ways to categorize and value land and property for taxation, but many countries adopt a market value and comparable sales approach [23]. However, this assumes that there is a vibrant land market operational and there are real estate experts and property taxation experts. However, in many developing countries this is not the case and Local Governments miss out on this crucial source of income. A FFP valuation approach being tested by the World Bank [23] involves extracting building characteristics from drone and street-level imagery using Machine Learning (ML) techniques to support a basic model (Minimum Viable Product) for land and property valuation. The information extracted from the aerial and street-level images includes land coverage and use, and building location, size, height, number of stories, building materials of roof and walls, windows and doors. Billboards and other signs can be used to derive the use of properties. This limited set of measurable tax value parameters can be efficiently maintained, is transparent and can be easily understood by all the stakeholders.

One of the cities used to test this approach by the World Bank's Global Program for Resilient Housing was a 7 km$^2$ section of Bogota in Colombia. Aerial images were collected using an 'ebee' drone with Post-Processed Kinematic (PPK) and base stations. Street view images were collected approximately every 2 m using a high-resolution (30 megapixel) car-mounted 360-degree or street view camera. Each image (JPG) file was delivered so that it included the "captured at" timestamp and corresponding WGS84 latitude and longitude coordinates. Metadata included the camera facing direction as actually recorded or systematically derived by an interpolation method and were capable of being stored in the image EXIF standard. The survey team used a Trimble MX7 with a Ladybug camera nestled inside a Trimble mobile mapping system. This portable system took 6 pictures every few seconds for a panoramic photo of 30 megapixels. This apparatus also allowed for an increase in geospatial accuracy. The Trimble MX7 system used tightly coupled GNSS and an inertial referencing system. In addition, it was also equipped with GNSS Azimuth Measurement Subsystem (GAMS) to continuously calibrate the inertial measurement unit (IMU) and ensure that azimuth did not drift—maintaining heading accuracy. The following lessons learned were derived from the project:

Advantages:

- A horizontal accuracy of 5–10 cms and a vertical accuracy of 10–20 cms were achieved for the dataset over this urban area.
- Many different types of building objects and characteristics were extracted and registered to support the valuation process.
- The minimal set of tax value parameters is easy to explain and creates a just and transparent taxation process.
- The ML process can be less expensive (particularly at scale), more flexible and faster than the traditional field surveying approach. Due to the low data acquisition costs,

the approach allows for more frequent data collection/maintenance cycles to be established, thus helping to build trust in the building valuations with the citizens.

- The tax value parameters are easy to maintain and can be augmented to support a more sophistication of the valuation model over time when there is demand.
- Standard Geographic Information System (GIS) tools can be used to process, categorize and visualize the data (Figure 2).
- Usually, there is a considerable lag in time between the completion of land registration and cadastral projects and corresponding property valuation projects. In the above scenario, it is feasible to combine the projects and use the same data capture techniques and share the data. This will allow property tax revenues to be brought on-line more quickly.

Disadvantages:

- The project area included properties with rooftops that were very tightly packed together, and the resulting ML polygons sometimes required manual disaggregation.
- The details of the owners/occupiers still require visits to the properties to carry out a survey. However, this can be an integral part of community meetings where the building objects extracted by ML can be verified.
- The capture of the drone and street level imagery requires good ground control or RTK with good access to the GNSS network. This can often be interrupted by foliage and buildings.
- The approach is markedly different from conventional approaches to valuation and professionals will need to be convinced of its adoption.

### 4.1.1. Check 1: FFP Approach Compliance

The FFP criteria (Table 1) was used to assess whether this case study meets the FFP kitemark and the following checks were carried out (Table 5):

**Table 5.** FFP Approach Compliance Check.

| | |
|---|---|
| **Flexible** | The types of land and building features extracted from the imagery can be modified through ML to support regional variation in building characteristics and a variety of simple valuation models. |
| **Inclusive** | The approach involves obtaining drone and street level imagery for all land and buildings in a city, including informal settlements where motorcycles are used in place of cars to obtain street level imagery. In addition, the valuation model is simple and easy to understand by all citizens. |
| **Participatory** | The capture process is not participatory, but all citizens and businesses are able to view and understand the land and building features used to calculate their valuation and tax. |
| **Affordable** | The costs of capturing and processing the drone and street level imagery are low in comparison to the formal aerial photogrammetry, ground surveying and manual creation of a building cadastre. |
| **Reliable** | The land and building features extracted from the imagery are obtained through ML and the algorithms ensure consistency and reliability. In addition, the data is derived quickly after the capture of the imagery, ensuring the data are up to date. |
| **Attainable** | The World Bank experience in building local capacity to manage these types of projects and use the associated toolkits indicates that it is not onerous and can be sustained with local resources tapping into remote technical support. |
| **Upgradeable** | When the land market matures then the simple approach to valuation modeling can be replaced with a market value or comparable sales approach, for example. |

This valuation, land administrative service complies with the FFP criteria and is clearly a cost effective and sustainable solution that can be achieved in a relatively short timeframe and merits the FFP kitemark.

### 4.1.2. Check 2: Common Supporting Data Compliance

The data captured to support the valuation case study exactly matches the data elements outlined in Table 2. In this case, the point cloud was derived from SfM rather than using a LiDAR sensor. This FFPLA valuation service therefore uses a common set of geospatial data that can be captured once and shared.

### 4.2. Case Study 2: Resilient Housing

In the context of rapid global urbanization, it is estimated that over the next 15 years the number of people living in substandard housing will double to 3 billion. Disasters, such as hurricanes, floods and earthquakes, are increasing and significantly threaten these homes. Access to formal, affordable, and safe housing is not keeping pace with this extraordinary growth in urbanization. Sustainable communities can be built through making housing safe and resilient to disasters and help to protect lives and livelihoods. The World Bank has a Global Program for Resilient Housing that has developed a methodology 'to predict which houses are at risk of getting damaged by natural hazards, to identify which can be made safe before it is too late, and to connect them with government subsidies and private capital'. The goal of the program is 'to help communities 'BuildBetterBefore' a disaster strikes and to save lives and protect livelihoods after one occurs' [51].

Urbanization across all countries around the world is estimated to increase in the following decades, but at varied rates. It is projected that 68 percent of the world's population will live in urban areas by 2050; an increase from 54 percent in 2016 [52]. India's small to medium sized towns are growing at a rate ranging between 48 to 185 percent [53]. This growth is outpacing the rate at which cities can respond to housing needs and in many countries this urban expansion manifests itself in slums that grow out of control. It is estimated that 900 million people currently live in slums and one in every four people will live in a slum by 2030 [54]. For many cities, just knowing where slums exist, and their condition is a difficult enough problem to solve before cities design interventions to mitigate the problem. However, when drone and street-level images are captured for neighborhoods, Machine Learning (ML) algorithms can be trained to extract specific characteristics of each property (such as façade and rooftop size, condition, material, density and regularity) allowing slums to be recognized. This fundamental information allows neighborhoods to be categorized within cities and interventions prioritized, especially in areas of risk.

An example of this approach is a World Bank resilient housing project in Colombia [55] that addresses the Colombian government's priority of creating resilient and inclusive housing; approximately 23 percent of all Colombian households currently live in substandard and inadequate housing. Quantitative improvements in housing directly reduce poverty and improve living standards. The COVID-19 crisis has highlighted even more the fundamental value of quality housing. Improved housing also mitigates disaster risks and encourages climate sustainability. Colombia ranks 10th globally in terms of economic risk posed by three or more hazards and has the highest recurrence of extreme events in South America [56]. To address this qualitative housing deficit, the government is introducing two housing subsidy programs, including one to incrementally improve housing. 'Experience has shown that under certain conditions it is more cost effective and life-saving to strengthen (retrofit) the existing housing stock and to construct robust buildings than it is to repair damaged buildings after a disaster: to Build Better Before. Studies have shown that every $1 in disaster mitigation saves from $4 to $10 in post-disaster reconstruction costs' [23]. The World Bank has piloted the use of drone and street level images (Figure 3) in combination with ML to identify buildings with specific construction material, size, and use characteristics to create information and monitoring systems for the optimization of housing subsidy allocation.

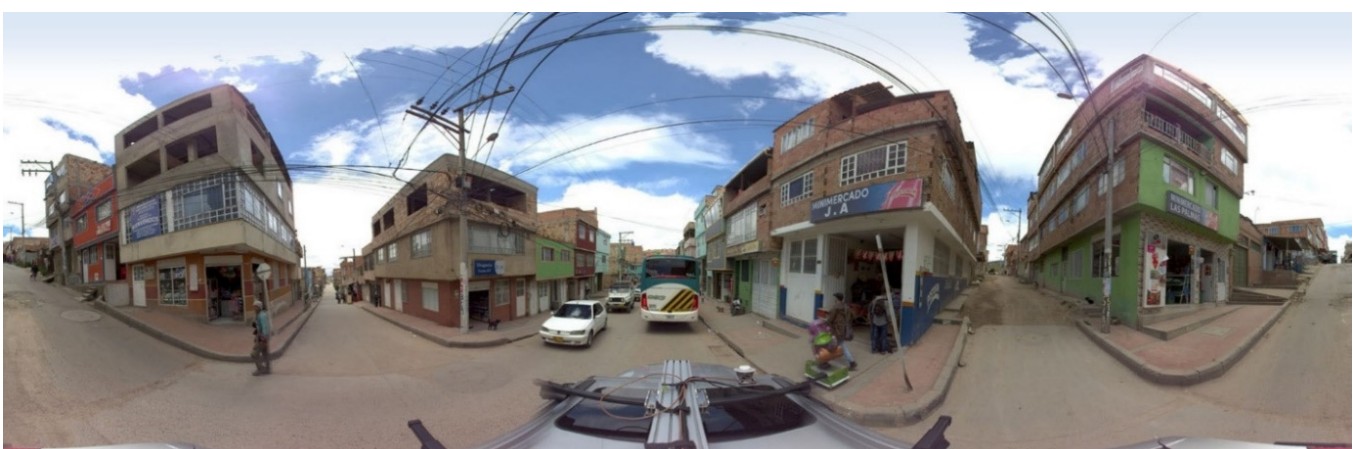

**Figure 3.** Street view panoramic from Colombia [source World Bank].

Advantages:

- Specific construction material, size, and use characteristics of properties were extracted to estimate the resilience of the property and to support the housing subsidy allocation process.
- The minimal set of resilience parameters was used to estimate the resilience of buildings. This made it easy to explain create a fair subsidy program.
- The ML process can be less expensive and faster than conducting a traditional survey approach, particularly at scale. The low acquisition costs allows data maintenance cycles to be established and ensures that the housing resilience model is current.
- If there is a need to change the housing resilience model then further property characteristics can be extracted from the imagery using ML.
- Standard Geographic Information System (GIS) tools can be used to process, categorize and visualize the data (Figure 2).
- This approach directly supports the 'BuildBetterBefore' approach being adopted as a priority for many governments and accelerates its implementation.

Disadvantages:

- The results of ML extraction of the building characteristics need to be verified through ground truthing.
- The capture of the drone and street level imagery requires good ground control or RTK with strong access to the GNSS network. This can often be interrupted by foliage and buildings.

### 4.2.1. Check 1: FFP Approach Compliance

The FFP criteria (Table 1) was used to assess whether this case study meets the FFP kitemark and the following checks have been carried out (Table 6):

**Table 6.** FFP Approach Compliance Check.

| | |
|---|---|
| **Flexible** | The types of building features extracted from the imagery can be modified through ML to analyze the resilience of buildings to different types of disasters, e.g., flood, earthquake, hurricanes. |
| **Inclusive** | The approach involves obtaining drone and street level imagery for all buildings in a city, including informal settlements. This supports the formulation of strategies for entire cities rather than just regions of the city. |
| **Participatory** | The capture process is not participatory, but the citizens and businesses in most need of improving the resilience of their buildings will be targeted for housing subsidy. |

| | |
|---|---|
| **Affordable:** | The costs of capturing and processing the drone and street level imagery are low in comparison to manually creating an inventory of all buildings in a city. |
| **Reliable** | The building features extracted from the imagery are obtained through ML and the algorithms ensure consistency and reliability. The building feature data are derived quickly after the capture of the imagery, ensuring the data are up to date. |
| **Attainable** | The World Bank experience in building local capacity to manage these types of projects and use the toolkits indicates that it can be sustained with local resources tapping into remote technical support. |
| **Upgradeable** | The model and the building features used to evaluate building resilience can be upgraded over time and extracted with ML. |

This housing resilience, land management function complies with the FFP criteria and is an affordable and easily maintained solution that can be achieved quickly and merits the FFP kitemark. A further example of this effective land management function is described below.

### 4.2.2. Check 2: Common Supporting Data Compliance

The data captured to support the housing resilience case study exactly matches the data elements outlined in Table 2. Although in this case, the point cloud was derived from SfM rather than using a LiDAR sensor and thus has a lower density. This FFP housing resilience service to support the optimization of housing subsidy allocation therefore uses a common set of geospatial data that can be captured once and shared.

### 4.3. Case Study 3: Solid Waste Management

It is estimated that the annual global damage of plastic pollution to marine ecosystems exceeds USD 13 billion. In the Asia-Pacific Economic Cooperation (APEC) alone, this causes over USD 1.3 billion per year to be lost to the tourism, fishing, and shipping industries [57]. A World Bank project in Cambodia identified that plastic waste pollution forms a particularly crucial part of solid waste mismanagement. The Mekong River is amongst the most polluted rivers worldwide and in Phnom Penh alone, about 10 million plastic bags are used every day [58]. However, dependable statistics on the varieties and quantities of plastic in the Cambodian rivers and sea are not available. These statistics are essential to establish robust policies and investment strategies to reduce plastic waste and prevent damage to the environment and the economy.

The World Bank's Cambodian project [59] piloted an innovative approach to monitoring and gathering these essential statistics. An RGB camera was mounted on a drone and images of polluted rivers, beaches, and urban canals were collected at different flying heights. The images were then analyzed using Machine Learning (ML) (Convolutional Neural Networks), and the automated image analysis enabled a wide range of the plastic pollution characteristics to be derived, including the detection of pollution hotspots, the size of areas covered with plastic, volume estimation, and importantly, classification of more than 10 different types of plastics. This remote sensing approach was combined with scientific field surveys to ground truth and improve the process of classification. A second phase of piloting is planned, and this will involve the use of a multispectral sensor to improve the overall data quality in terms of detection and classification of plastic waste.

The introduction of solid waste management policies requires significant investment and a key, sustainable source of this financing is normally achieved by municipalities through collecting waste fees from households. This requires the registration of properties and the creation of street/property gazetteers to underpin the fee collection system. This geospatial and property information can then be re-used by municipalities to support a range of their core activities, including the collection of property taxes, spatial/urban planning as well as environmental resilience planning [58]. In this Cambodian example, the introduction of solid waste management is the policy driver that is generating the base

data identified in this study to support a wide range of other FFP land administration and management applications.

Advantages:

- A wide range of the plastic pollution characteristics was derived to help formulate a more effective set of solid waste management policies.
- These inputs to policy formulation through the ML process are much cheaper and faster than conducting a traditional survey approach.
- The ability to identify and classify more than 10 specific, different types of plastics allows a much more targeted and effective anti-pollution policy to be formulated and implemented.
- The on-going monitoring of the plastic pollution allows the impact and effectiveness of the policies to be measured.
- The level of resolution of the drone imagery required for the ML process is high to allow the identification of specific plastic pollutants. This is fully compatible with the resolution required for FFP land tenure applications.
- To attempt to achieve the monitoring and classification of plastics in rivers by manual means would be problematic, resource intensive and expensive, normally beyond the means of the local environmental agencies involved. This FFP approach provides a very low-cost solution and delivers significant benefits through more effective policies to reduce the level of plastic waste.

Disadvantages:

- The results of ML extraction of the plastic characteristics need to be verified through ground truthing, although the future use of a multispectral sensor will improve the overall data quality.
- This application does not need to have a high degree of positional accuracy when the drone imagery is georeferenced. To impose this requirement on capturing the drone imagery with good ground control or RTK would impose overheads on the project.
- Properties need to be registered and a street/property gazetteer created to underpin the fee collection system. However, the gazetteer can have many other uses within urban management, including property tax revenues.

4.3.1. Check 1: FFP Approach Compliance

The FFP criteria (Table 1) was used to assess whether this case study meets the FFP kitemark and the following checks have been carried out (Table 7):

**Table 7.** FFP Approach Compliance Check.

| | |
|---|---|
| **Flexible** | The types of waste material extracted and classified from the imagery can be modified through ML to be attuned with the types of pollution occurring in the area of interest. |
| **Inclusive** | The monitoring of pollution can be performed across an entire city or just known hotspots. |
| **Participatory** | Communities could be involved in ground truthing the results of ML. |
| **Affordable** | The costs of capturing and processing the drone are low in comparison to manually creating an inventory of pollutants. |
| **Reliable** | The waste material features extracted from the imagery are obtained through ML and the algorithms ensure consistency and reliability. |
| **Attainable** | The World Bank experience in building local capacity to manage these types of projects and use the toolkits indicates that it can be sustained with local resources tapping into remote technical support. |
| **Upgradeable** | The pollutant types used to formulate solid waste management policies can be upgraded over time and extracted with ML. |

This solid waste management land management function complies with the FFP criteria and is affordable, provides ongoing monitoring of the impact of policies and can be achieved quickly and merits the FFP kitemark.

### 4.3.2. Check 2: Common Supporting Data Compliance

The data captured to support the solid waste management case study does not fully match the data elements outlined in Table 2. The georeferencing of the drone imagery is less rigorous than that required for FFP land tenure and there is less of a requirement to create sufficient overlaps of the imagery to support 3D photogrammetric models. To apply these more stringent requirements on the data capture process will increase the complexity and costs of the project. However, these costs could be shared across a number of other projects using the data. This FFP solid waste management service to support the more targeted solid waste management policies can use a common set of geospatial data. However, this will impose higher costs on the project and required more skilled personnel.

### 4.4. Other FFP Land Management Function Opportunities

The generation of drone and street level imagery along with the derived data outlined in the FFP land administration services and FFP land management functions above, provide opportunities to support a wide variety of derivative, innovative urban land management functions. Examples of these are provided below.

### 4.4.1. Master Planning

The UN estimates that India currently accounts for 11 percent of the world's population and this will increase to around 15 percent by 2030. The Census of India has identified that the urban population has more than double from 14 percent at the time of Independence to around 32 percent in 2011 and the number of cities/towns has increased from 5161 to 7933 over this period. Urbanization will increase to more than 50 percent by 2051. India has been finding it increasingly difficult to plan the development of its cities. The majority of urban settlements in India can be considered unplanned with uncontrolled growth leading to loss of agricultural land in peri-urban areas, large numbers of informal settlements, significant environmental damage, and a poor quality of life for its citizens [53].

The Master Plan/Development Plan is a fundamental element of urban land management and facilitates sustainable development through land use allocation. These Master/Development plans are designed to manage development over a 20-year period with periodic reviews and revisions every five years. These plans require base mapping to be prepared to support the collection of existing land use surveys and socio-economic data. However, less than a third of the over 8000 towns and cities have master/development plans. The main reasons for this backlog are the lack of quality geospatial data along with lack of capacity and funding in small to medium sized towns that make up 93 percent of the towns and cities. The master planning of these small and medium sized towns is a priority for the Indian government as they are growing at a rate ranging between 48 and 185 percent. The development of these urban areas would support a reduction in the considerable migration to large cities and help to sustain growth in the surrounding rural areas [53].

To accelerate the master planning process across small and medium sized towns, the Ministry of Housing and Urban Affairs (MoHUA) has decided to promote the use of drones to generate the base maps required for the master planning process. This cheaper alternative to manned aerial photography will unlock this capability. The MoHUA has produced a draft set of process designs and standards to use drones in the generation of 1:1000 standardized mapping. This was required to support geospatial data of high accuracy and better spectral and spatial resolution at low cost as well as the 3D models derived from the sensor data provided by drones. In addition, the new standards supported more effective interoperability and seamless vertical integration from Local to Regional Plans These specifications will be tested in ten towns during 2021 before being scaled up

across India [53]. This drone-based initiative in India, primarily focused on supporting master planning, opens up significant opportunities to re-use the drone data to support a wide range of land administration services and land management functions highlighted in this paper. For example, Machine Learning could be used to support change detection crucial in modeling city growth patterns.

### 4.4.2. Urban Digital Twins

Digital twins are a digital, virtual representation of reality and over the past decade have been used to create 3D, often real-time, representations of cities. Singapore, Glasgow, Boston, Jeonju and Jaipur were the early adopters. It is projected that by 2025, the number of urban digital twins globally will be over 500 and the market will have grown from USD 3.8 billion in 2019 to USD 35.8 billion per year [59]. These integrated and comprehensive geospatial information systems are given the accolade of providing the continuous innovation needed to construct new smart cities.

Urban digital twins have a wide set of applications; utilities can plan infrastructure coverage; environment managers can conduct scenario analyses through the simulation of the potential impact of natural disasters like flooding; telecommunications companies can more effectively plan their dense networks of 5G antennae in 3D; urban planners can perform better land use analysis and engage with citizens through immersive visualization experiences with AR/VR applications; the car industry is designing navigation solutions for autonomous vehicles; green energy providers can optimize energy savings and solar capacity; emergency responders can conduct disaster readiness simulations to build city resilience; and the entertainment industry can build seamless computer-generated imagery integration [60]. The result is much more resilient and efficient cities—smart cities.

These 3D urban digital twins are created from multiple sources, including optical and satellite imagery, LiDAR, IoT and crowdsourcing from the citizens. The data created by the use of the emerging, innovative technology being applied to FFP land tenure can be directly used to build these urban digital twins and accelerate the creation of smart cities.

### 4.4.3. Visualizations to Support Gaming

The point clouds and 3D models generated by this innovative technology are also being adopted by the gaming sector to create more realistic, virtual worlds for enhancing the gaming experience. Ordnance Survey GB have been involved in an initiative to use the OS OpenData products to create a Minecraft world representing over 224,000 km$^2$ of Great Britain [61]. HERE Technologies has released high-fidelity, 3D models of 75 city centers from around the world to give software developers the geospatial data required: data from over 100,000 sources and with 80 billion API calls per month. This provides the entertainment industry with seamless computer-generated imagery integration to build real-world visualizations of cities [60].

### 4.4.4. Support of Financial Risk Prediction

The base datasets being derived from FFP land tenure also have the potential to be used commercially. This could generate revenue to help finance the implementation and maintenance of land administration and land management applications. For example, street level imagery is finding increasingly innovative uses. Researchers [62] have used images of residences from Google Street View to predict the likelihood that a policyholder would make a car accident claim. Residences are classified using the type (detached house, terraced house, block of flats, etc.), age, and condition extracted from the imagery and the characteristics used to model and predict the likelihood of a car insurance claim. The correlation was found to be surprisingly strong and when these residence factors were included in the insurer's risk model, the predictive power was increased by 2 percent. This use of residences could equally be applied to improve predicting risk for insurance companies, financial services, and health-care organizations [62]. This use of street level imagery raises important issues around data privacy laws and whether clients' consent

given to a company to store their addresses also provides consent to store information about the appearance of their houses. Many countries, including Germany and Austria, have raised privacy concerns over Google's Street View product and coverage has stopped in India [63]. Users of street level imagery need to be aware of the potential limitations imposed through licensing restrictions and privacy laws.

### 4.4.5. Disaster Risk Management

There were 28 million new displacements associated with conflict (10.8 M) and disasters (17.2 M), including hurricanes, droughts and floods, across 148 countries and territories in 2018 [64], and there is now growing evidence that the impact of natural disasters and associated vulnerability are intensifying. This influence is seriously challenging development and the goal of achieving the SDGs [65]. Countries must implement disaster risk reduction and resilience building measures, rather than just respond to one-off disaster events [65] and provide communities with greater support to adapt to these demanding environmental conditions.

Disaster Risk Reduction is 'the concept and practice of reducing disaster risks through systematic efforts to analyze and manage the causal factors of disasters, including through reduced exposure to hazards, lessened vulnerability of people and property, wise management of land and the environment, and improved preparedness for adverse events' [66]. The management of disaster risk involves a whole range of phases involved in the lifecycles of disasters, including disaster prediction (simulation and visualization), prevention, preparedness and mitigation, emergency response, evacuation planning, search and rescue, shelter operations, and post-disaster restoration and monitoring [67]. Fundamental to managing all of these phases is access to geospatial data and information contained within land administration services. For example, prediction through flood modelling requires DEM and DTM information, the restitution of properties following a disaster requires land registration and cadastral information to re-establish security of tenure for citizens, and financial compensation schemes need valuation information about the impacted land and properties. Without this fundamental information, the ability to manage disaster risk is seriously reduced.

The baseline geospatial data, a key part of the Minimum Viable Product identified in this study, can be captured efficiently and cheaply to derive the information need for land administration and management that can then be used to manage resilience and disaster risks. The recovery from the effects of a hazard can subsequently be achieved more quickly and efficiently, allowing essential basic structures and functions to be restored. The drone and streel level imagery alone are a valuable information asset to support many phases of the disaster management process.

These geospatial and land information services records and data need to be safeguarded to avoid any physical destruction and the information needs to be digitalized and critically, good practice data management practices applied.

## 5. Discussion

These case studies reveal that the FFP paradigm that has predominantly been applied to the land tenure aspects of land administration has been successfully adopted and implemented in wider land administration services and land management functions. A key element of the FFP approach is the use of a well-defined Minimum Viable Product as an entry level solution that meets the requirements of stakeholders, is easily understood, affordable and fast to implement. The Valuation and Property Tax Revenues case study is an excellent example of how to apply a Minimum Viable Product [23]. The adopted Minimum Viable Product valuation model is simple, easily understood, efficiently captured and maintain through Machine Learning (ML), and can be upgraded when required. Solutions should not be driven indiscriminately by technology, but should be primarily guided by user needs. However, in the case of these emerging FFP solutions, the innovative technologies described in this study have formed a symbiotic relationship with the FFP

paradigm and are creating opportunities for implementing new FFP approaches that would not otherwise be feasible.

The use of these innovative technologies requires specialized skills and this capacity to effectively use and maintain the technology is not normally found within the governments most in need of adopting the technology. However, the World Bank's approach is to provide management skills and guidance to governments to allow the effective management of partners in the private and academic sectors to provide these sustainable services. This is providing opportunities for private sector companies to build this specific capacity.

The analysis of the emerging technologies and data used to support the FFP case study solutions has revealed that there is a common set of geospatial datasets that can be captured once and shared across many other land administration services and land management functions in an urban environment. FFP land tenure solutions capture data with the most demanding specifications and when these data are augmented with street level imagery then the combined data portfolio supports a surprisingly wide range of uses. This study has identified support for a range of land management functions, including urban digital twins, master planning, disaster risk management and financial risk prediction. So, although there are financial and resource overheads associated with data capture in the initial FFP solution implemented in an urban environment, the return on investment is significant since many more FFP solutions can be implemented using this shared portfolio of datasets.

Most land tenure programs simply focus on securing land rights for citizens, but this is just the start of their journey to leverage the economic potential of their land and property. For example, smallholder farmers (there are an estimated 500 million smallholder farmer families worldwide [68]) also require access to technical and financial services to improve their livelihoods. The incremental and fragmented delivery of these services makes their sustainable prosperity much more challenging. Therefore, an integrated package of technical and financial services, including security of tenure, would achieve much higher benefits from a portfolio of interventions. However, donors and Non-Governmental Organizations (NGOs) find this more holistic approach difficult to adopt due to their organizational divisions, silos of professional skills, difficulty in managing multi-faceted programs and complexity of interfacing with a range of recipient government ministries and departments. None of these current perceived restrictions are showstoppers, but adopting new approaches will require significant cultural and institutional change.

The FFP paradigm and the emerging, enabling technologies provide compelling opportunities to rethink how land administration and land management programs are designed, integrated and implemented. This research has identified that a common set of base geospatial data, collected by and derived from drone and street level imagery, can support a wide set of FFP land administration services and management functions, ranging from land registration and cadastre to valuation and urban resilience. Therefore, it is feasible that land intervention programs, initially involved in a single land administration or land management activity within a region, could consider capturing this identified base geospatial data to a threshold of quality that would then also support a wide variety of complementary land administration services and land management functions. This more holistic approach would provide the opportunity for single land interventions projects to be integrated into a wider program of land administration services and land management functions delivering a more significant set of socio-economic benefits. For example, a land and property valuation project could be implemented in parallel with other projects to help finance this wider set of projects through property-based tax revenues. Although FFP land tenure projects deliver a fundamental framework of land rights information that normally supports subsequent land administration and land management projects, it is still feasible to reverse this order and build FFP land tenure projects on the back of other land projects.

This integrated approach to program implementation has significant implications for the future design of FFPLA projects where there will be opportunities to deliver an integrated portfolio of land administration services and land management functions more quickly and efficiently compared to the traditional approach of delivering independent, single projects.

Too often, land administration projects are provided with the capital budget to implement the project, but lack the revenue budgets, resources and processes for on-going maintenance of data underlying the services. An example of this situation occurred in Rwanda where, after the very successful implementation of a national FFPLA project in just five years, subsequent poor maintenance of the land rights jeopardized the original investment [69]. Since the FFP approach involves defining a Minimum Viable Product [9] as a starting point, the corresponding data models to support the FFP projects are much simpler with the range and complexity of the data limited to meet initial needs. This makes the maintenance of the data less onerous and more achievable with fewer resources. A further advantage is achieved with the proposed integrated FFP approach where a common set of baseline data supports many FFP land administration services and land management functions. In addition, many of the data are semi-automatically extracted from the imagery using ML, leading to even more efficient data capture for maintenance.

The FFPLA concept is composed of three frameworks: spatial, legal and institutional. This study has primarily focused on the spatial framework and identified how emerging technologies are facilitating the wider use of the FFP approach in land management functions. However, this wider adoption is dependent upon similarly supportive legal and institutional frameworks. For example, how can institutions be encouraged or mandated to share both geospatial datasets and the costs of their collection and maintenance? Does the legal and regulatory framework within a country need to be adapted to accommodate these new FFP approaches, including valuation models, data quality specifications and the move from national mapping to more autonomous city mapping? How can the current project siloes be dismantled to be replaced by integrated programs sharing data, costs and resources? Would the results of a socio-economic impact assessment of implementing a FFP approach deliver a positive return on investment in comparison to conventional approaches and thus convince politicians to invest? Further research is required on how legal and institutional frameworks should be adapted to be more compliant with this new vision.

## 6. Conclusions

FFPLA, in particular FFP land tenure, is gaining acceptance around the world as a fundamental methodology to scale up the provision of security of tenure [5–8] and hopefully coalesce the currently excluded 70 percent of the world's population into formal land administration systems within a generation. There are many aspects of the FFP methodology that support scalability, including participatory, flexible, inclusive and affordable. However, emerging innovative technologies [14] are enablers and will continue to significantly accelerate the implementation of the FFP methodology. Drone and street level imagery are affordable, require less professional support to operate, can adequately cover small and medium sized towns, provide the required resolution and accuracy to support cadastral applications, and point clouds can be derived or directly captured with LiDAR. Orthophotos can then be directly used in the field in a participatory process with the citizens to identify and validate visible parcel boundaries. Machine Learning can increasingly be used to automatically extract parcel boundaries from the imagery, with support from the point clouds, to aid in the parcel boundary validation process with citizens.

The base data captured for FFP land tenure services can be augmented with street level imagery and then be directly re-used to support a much wider set of FFP land administration services and land management functions. For example, building features and characteristics can be automatically extracted from the drone and street level imagery using Machine Learning to investigate aspects of city resilience and to better target funds to mitigate risks and improve resilience. The data can also be used to create urban digital twins that support a range of opportunities for the sustainable management of urban environments.

The base data also provide opportunities to generate revenue to support and accelerate the implementation of land administration services and land management functions. For example, Machine Learning derived building characteristics can support land and property valuation and the downstream generation of property tax revenues, and images of properties could be sold to insurance companies, financial services, and health-care organizations to improve risk prediction.

The ability to directly re-use the identified base data across a wide range of FPP land administration services and land management functions will allow international funding institutions and development agencies involved in financing land initiatives to reconsider how their land related programs are sequenced, integrated and financed. Traditionally, land tenure projects are prioritized followed sequentially by further land administration services and land management functions. However, the sharing of the base data across a number of FFP solutions can support the parallel implementation of these FFP solutions. This will accelerate the implementation, reduce the associated costs and support more effective maintenance of the data. A more holistic approach to administering and managing land through the delivery of an integrated portfolio of land interventions will leverage the economic potential of land and property more effectively and quickly. This new method will impact the approach to land governance and the institutional arrangements that deliver sustainably managed land, property and natural resources. These alterations should be reflected in analytical tools, such as the World Bank's Land Governance Assessment Framework (LGAF) [70], to encourage these transitions. However, this will require a significant cultural, professional and institutional change from all stakeholders involved. The administration and management of land is normally highly fragmented across a wide number of land professions in specific niches. However, the wider adoption of this FFP approach provides an opportunity for the land professionals to reconsider their professional domains and re-align their professional roles in a more effective, integrated network of services. Although challenging, it is a great opportunity that must be embraced to ensure that land is sustainably managed and delivers significant value to societies and their economies.

**Author Contributions:** K.K.: visualization; conceptualization; validation; investigation; writing—review and editing. S.A.: visualization; conceptualization; validation; methodology; investigation; writing—review and editing. R.M.: visualization; conceptualization; methodology; investigation; formal analysis; writing—original draft preparation. All authors have read and agreed to the published version of the manuscript.

**Funding:** This research received no external funding.

**Institutional Review Board Statement:** Not applicable. All the material used in the study has already been published in World Bank Group reports and no new studies were undertaken.

**Data Availability Statement:** All the material is referenced in the manuscript and is available on the web.

**Acknowledgments:** The authors would like to thank Stig Enemark for encouraging the publication of this paper, Simon Wills for providing help with the references, and William Mackaness for providing excellent guidance.

**Conflicts of Interest:** The authors declare no conflict of interest.

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
