# Peer review of "Applying the FFP Approach to Wider Land Management Functions"

_land, doi:10.3390/land10070723_

Round 1

Reviewer 1 Report

This is an excellent contribution to the literature drawing on World Bank project experiences as well as literature. It is well written and draws on appropriate literature.

Section 2 approach and methodology - while clear about the approach of using the seven elements to evaluate the case studies to see if they are FFP and support sustainable development goals, more information is needed on this. For example, Section 2 should mention that this study is focussed primarily on the FFP spatial framework as mentioned in the discussion.

Footnotes used in the manuscript should be converted to citations.

Author Response

Thank you very much for your constructive review of the article.

Reviewer 2 Report

This is an excellent paper, well written and full of interesting observations. The authors are well experienced in the fields of land administration and are able to link a variety of threads together in the FFP construction of what used to be called a land information system or multi-purpose cadastre. This paper is well worth publishing.

Author Response

Thank you very much for your positive comments on the article.

Reviewer 3 Report

This manuscript does an excellent job of describing a wider set of applications for Fit-for-Purpose land administration technology and presenting interesting case studies as evidence.  

The manuscript could be made more useful and generalizable for the overall community of practice utilizing FfP, including low-income countries and fragile states, with minor revisions in the following three areas: 

  1. The manuscripts examples are heavy on World Bank-supported cases (e.g. lines 113-118, line 358, case studies).  The argument about the advantages of these new technologies would be enhanced if the article could cite additional examples in which the FfP technologies are being applied for these wider applications without the financial support of a multi-lateral development bank, especially examples from low-income countries and fragile states. The article would read as more neutral and more general with less emphasis on the WB's role in promoting these technologies. 
  2. Add a description of the basic requirements for user capacity and long-term system maintenance for each technology. Presumably these requirements also compare favorably to traditional/conventional technologies. 
  3. Add a specific cost comparison to support the "Affordable" criterion in Tables 4, 6 and 8, perhaps as a paragraph in the text footnote.

Author Response

(The authors gave the same response as above.)

Reviewer 4 Report

Wide-ranging article, well structured, referenced and presented, but perhaps tries to cover too much ground.  I only have minor suggestions:

1) Acronyms frequently used, sometimes may not be familiar to readers, and their meaning may be forgotten, even though given at first occurrence. Examples: ML is only used a few times, so full version would help understanding, likewise MVP is mentioned early on, but does not re-appear until much later. Some acronyms seem to be internal to WB (to use one acronym!), less familiar outside, and can quickly become jargon understandable to the cognoscenti). Think about accessibility to intended readership.

2) Section 4 is an example ('FFP land management function opportunities', another example of WB jargon). Suggest simplifying each sub-section in 4 to a paragraph or two each.

3) Conclusions could be longer, and relate more to the governance challenges in getting the approach accepted. Reference to WB LGAF (land governance assessment frameworks) could help. There is mention of 'cultural, professional and institutional change from stakeholders', but how is this to be achieved. Likewise, 'more holistic land administration and management approach' deserves further discussion (sounds like a FIG take-over bid).

Author Response

Thank you very much for your constructive review of the article.

This manuscript is a resubmission of an earlier submission. The following is a list of the peer review reports and author responses from that submission.

Round 1

Reviewer 1 Report

The article deals with a very interesting subject. Properly reworked, it will interest many readers. However, my main reservation is that, as it stands, it is more of an extremely good expert opinion piece than a typical scientific article. Therefore, if the authors were interested in a scientific publication, they should make the following changes:
1. many parts (e.g. subsections 1.3, 2.1) lack in-depth reference to the theory and literature of the subject. Such developed considerations are a necessary part of scientific articles. And the review part should be (besides the Discussion) one of the most important parts of the article. In my opinion, it is worth seriously expanding it, and perhaps even preparing it anew.
2. the layout of the text from page 6 to the Discussion should be made more readable. There should be better justification of why particular issues are discussed (justification also from a theoretical perspective). This section should be shortened.
3. the current Discussion does not, in my opinion, meet the standards for Discussions in scientific journals. The Discussion should very strongly refer to the Review, to the theses that were referenced in the Review. The Discussion should analyse these scientific theses in more detail, in the context of the obtained results. It is not, therefore, about containing a general set of recommendations. In my opinion the Discussion should be rewritten.
In order to make the authors' work easier, I propose that they indicate even more clearly what the scientific novelty of their article is. In the Conclusions, they should indicate further research directions.

Dear authors, I highly appreciate your preparation for the topic. I am impressed by your experience and freedom in making recommendations. My comments are only intended to adapt your excellent postulates and thoughts to the scientific framework. I am confident that you will succeed in doing so. Good luck.

Reviewer 2 Report

Review of the article: Applying the FFP Approach to Wider Land Management Functions

Introduction: „This paper explores how the FFP methodology can be applied to a wider set of land administration services and land management functions. By the authors the article also discusses: the innovative use of these emerging technologies and identifies a common set of data capture techniques and geospatial data that can be shared across a range of urban land administration and management activities. Finally, the paper discusses how individual land projects could be integrated into a more holistic land administration and management program approach and deliver a significant set of socio-economic benefits more quickly.

  1. Introduction (subsections 1.1 – 1.3) - clearly introduce the subject.
  2. Approach and Methodology - Approach - ok; Methodology - no description of the methods used (there is only the purpose of the study).
  3. Innovative technologies to support FFP are described - types of drones, remote sensing techniques, ground photos (in combination with geo-referencing), application of ML are described - even interesting.
  4. FFP Assessment of Wider Land Administration and Management Opportunities - In 4.1 - Valuation and Property Tax Revenues - the FFP criteria checks have been carried out; 4.2 - et cetera… examples of use and control of FTP criteria are described.
  5. Discussion - quite modest, with such a wide description of applications

The article is descriptive (overview) - description and examples of applications of digital representations of reality. But even interestingly written.  It is difficult to talk about research methodology here (which I scored earlier). Even interesting examples from the world are described.

Reviewer 3 Report

The abstract must be changed. The introductory part on the importance of FFP should be reduced.

Within the abstract, in addition to the presentation of the objectives, a series of obtained results should be presented.

The introductory part is very well done, highlights, based on the bibliography, the importance of this study. The part reserved for the presentation of UAV (drone) is too developed, it is possible to give up the purely technical aspects such as the presentation of RTK, cloud point, etc.

Chapter 2 begins with the objectives of the study and further presents the context of the study, this introductory part cannot be part of the methodology (it can be introduced in the introductory part). In chapter 2, being a methodological chapter, it is necessary to insist concretely on the methodology followed and / or developed for the realization of the proposed research.

Chapter 2 is more of an Approach chapter than a methodological one. The chapter does not present a clear proposed methodology or an adapted methodology or a derived methodology. There are no methodological stages, sub-stages, transition links between them ....

The entire chapter 3 presents technologies that can be implemented in the proposed research. All these are presented at a theoretical level, based on bibliographic sources, there is no basis on their concrete application in the study presented. for example, the drones are presented, highlighting their classification, the types of sensors used, LiDAR and the main output databases are also presented at a theoretical level ........ All this information is known, the technical implementation and their place in the presented methodology should be highlighted.

The rest of the article continues in the same direction, theoretically, based on bibliography without analyzing a concrete study that will highlight problems or advantages of the methodology.

The whole study should be restructured, it should insist on a case study through which to highlight the methodology and to highlight the implementation of modern techniques, advantages and disadvantages of implementing these techniques .....